# Cervical lesion proportion measure using a digital gridded imaging technique to assess cervical pathology in women with genital schistosomiasis

**Louise Thomsen Schmidt Arenholt**[1,2,3], **Katrina Kaestel Aaroe**[1], **Kanutte Norderud**[1], **Mads Lumholdt**[1], **Bodo Sahondra Randrianasolo**[4], **Charles Emile Ramarokoto**[4], **Oliva Rabozakandraina**[4], **Dorthe Broennum**[1], **Hermann Feldmeier**[5], **Peter Derek Christian Leutscher** [1,3] *

**1** Centre for Clinical Research, North Denmark Regional Hospital, Hjoerring, Denmark, **2** Department of Gynecology and Obstetrics, North Denmark Regional Hospital, Hjoerring, Denmark, **3** Department of Clinical Medicine, Aalborg University, Denmark, **4** Association K'OLO VANONA, Antananarivo, Madagascar, **5** Institute of Microbiology and Infectious Diseases Immunology, Campus Benjamin Franklin, Charité—University Medicine Berlin, Corporate Member of Freie Universität Berlin, Humboldt-Universität zu Berlin and Berlin Institute of Health, Berlin, Germany

* p.leutscher@rn.dk

**Data Availability Statement:** Data are available in the RedCAP data management hosted by the

## Abstract

Female genital schistosomiasis (FGS) is characterized by a pattern of lesions which manifest at the cervix and the vagina, such as homogeneous and grainy sandy patches, rubbery papules in addition to neovascularization. A tool for quantification of the lesions is needed to improve FGS research and control programs. Hitherto, no tools are available to quantify clinical pathology at the cervix in a standardized and reproducible manner. This study aimed to develop and validate a cervical lesion proportion (CLP) measure for quantification of cervical pathology in FGS. A digital imaging technique was applied in which a grid containing 424 identical squares was positioned on high resolution digital images from the cervix of 70 women with FGS. CLP was measured for each image by observers counting the total number of squares containing at least one type of FGS associated lesion. For assessment of inter- and intra-observer reliability, three different observers measured CLP independently. In addition, a rubbery papule count (RPC) was determined in a similar manner. The intra-class correlation coefficient was 0.94 (excellent) for the CLP inter-rater reliability and 0.90 (good) for intra-rater reliability and the coefficients for the RPC were 0.88 and 0.80 (good), respectively. The CLP facilitated a reliable and reproducible quantification of FGS associated lesions of the cervix. In the future, grading of cervical pathology by CLP may provide insight into the natural course of schistosome egg-induced pathology of the cervix and may have a role in assessing praziquantel treatment efficacy against FGS.

**Trial Registration:** ClinicalTrials.gov, trial number NCT04115072; trial URL https://clinicaltrials.gov/ct2/show/NCT04115072?term=Female+genital+schistosomiasis+AND+Madagascar&draw=2&rank=1.

Centre for Clinical Reseach, North Denmark Regional Hospital, 9800 Hjørring, Denmark (phone: +45 97 64 26 52; mail: forskning.rhn@rn.dk - contactperson: Signe Westmark.

**Funding:** The study was funded by Merck KGaA (https://www.merckgroup.com/en). PDCL received the funding. The funder had no role in study design, data collection and analysis, decision to publish, or preparation of the manuscript.

**Competing interests:** The authors have declared that no competing interests exist.

## Author summary

Female genital schistosomiasis (FGS) is characterized by development of egg-induced chronic inflammatory lesions of the cervix and the vagina. FGS causes various symptoms such vaginal discharge, genital itch, pelvic pain and post-coital bleeding, and the disease is further associated with reproductive complications such as ectopic pregnancy and infertility. Moreover, FGS is today hypothesized as a risk factor for transmission of HIV in Sub-Saharan Africa. General prevention directed against *Schistosoma* infection and use of praziquantel as anthelmintic drug therapy are cornerstones in the FGS control strategy. The aim of the study was to test inter- and intra-rater reliability using a cervical lesion proportion (CLP) measure in a series of digital images from women living in a *Schistosoma haematobium* hyperendemic area in Madagascar". In that overall context, we have developed a digital image-based tool for quantitative assessment of FGS associated cervical lesions, which enables evaluation of treatment outcome at individual as well as community level with particular focus on resolution of cervical pathology, but also on risk of recurrence. The tool will also provide new information in understanding the natural history of FGS including development of clinical pathology.

## Introduction

*Schistosoma haematobium*, a trematode worm, occurs in Africa and in the Middle East [1]. In women *S. haematobium* affects particularly the external and internal genital organs in addition to the urinary tract. The adult worm pairs migrate through the venous plexus anastomoses in the pelvic region. Eggs are then trapped in the genital tissue evoking egg-induced inflammatory chronic changes with granuloma formation followed by fibrotic transformation. blood vessel distortion and calcifications of the cervico-vaginal epithelium [2]. These pathological manifestations occur in the disease entity of female genital schistosomiasis (FGS) [3]. Millions of women suffer from FGS, which causes a spectrum of genital signs and symptoms, including vaginal discharge, genital itch and pelvic discomfort [2–6] Moreover, FGS may lead to infertility and ectopic pregnancy [7,8]. In an overall context, FGS is associated with a reduced health-related quality of life and may cause stigmatization [9,10]. FGS is hypothesized to be contributing risk factor for transmission of human immunodeficiency virus (HIV) in *Schistoma* endemic areas of Sub-Saharan Africa [11,12].

A pocket atlas of the characteristic clinical pathology has been published by World Health Organization (WHO) to assist health-care workers in the diagnosis of FGS in *Schistosoma* endemic areas [13]. At the cervix, four types of lesions are associated with FGS: grainy sandy patches, yellow homogeneous sandy patches, rubbery papules and neovascularization [14,15]. According to a consensus recommendation, at least one of the four types of lesions should be present to establish the clinical diagnosis of FGS [16].

Furthermore, women should undergo colposcopy to ensure proper visualization and identification of the cervical lesions. This examination requires a colposcope, which is an expensive diagnostic instrument requiring maintenance and usually not available in schistosomiasis-endemic areas [17]. Moreover, colposcopes with an inbuilt digital camera are virtually non-existent. Thus, the interpretation of clinical pathology of the cervix is provided in a descriptive written form at best. Finally, proper use of a colposcope for diagnostic purposes requires extensive training. This makes comparison of changes in cervical pathology difficult. If the clinical pathology of the cervix is not available in a standard digital format, assessment of

diagnostic accuracy will be impossible. A digital method to document clinical pathology in women with FGS in a standardized and reproducible manner is needed. Such a method would be extremely useful in understanding the natural history of the pathology as well as to assess the efficacy of chemotherapy with praziquantel.

We applied a camera equipped with a macro-lens generating high-resolution digital images of the cervix. The images were then reviewed by use of a digital gridded imaging technique enabling observers to quantify lesions on the cervical portio. The aim of the study was to test inter- and intra-rater reliability using a cervical lesion proportion (CLP) measure in a series of digital images from women living in a *Schistosoma haematobium* hyperendemic area in Madagascar.

## Methods

### Ethics statement

The study was carried out in accordance with the Declaration of Helsinki and the Guideline for Good Clinical Practice. Approval was obtained from the Committee of Ethics at the Ministry of Health in Antananarivo (Comité d'Ethique de la Recherche Bio-Médicale auprès du Ministère de la Santé Publique), Madagascar. Informed written consent was obtained from all the participating women, including gynecological examination, specimen sampling and image-documentation of clinical pathology. A written consent was obtained from the parent/guardian of participants below 18 years of age.

### Study linkage and location

The gridded imaging technique study was performed in conjunction with a randomized controlled trial (RCT) in Madagascar assessing the efficacy of different dosages of praziquantel to reduce pathology of the cervix (ClinicalTrials.gov, trial number NCT04115072; trial URL https://clinicaltrials.gov/ct2/show/NCT04115072?term=Female+genital+schistosomiasis+AND+Madagascar&draw=2&rank=1). The trial participants lived in the district of Ambanja in the Northwest region of Madagascar, a highly endemic area for *S. haematobium* [18]. In total, 116 participants were recruited from two primary health centers in the municipalities of Antsakoamanondro and Antranokarany. Inclusion criteria for participation in the clinical trial were age 15 to 35 years and presence of cervical lesions associated to FGS. Baseline information on medical history and complaints was obtained from the study participants. Urine samples were collected, and 50 ml of urine were filtrated through a polycarbonate membrane [19]. After treatment at baseline with praziquantel the study participants were followed up after 5, 10 and 15 weeks involving a clinical re-assessment including photo documentation of the cervical pathology.

### Collection of digital images

A gynecological examination using a speculum was performed at baseline and at each follow-up visit. Photographic documentation of the cervix was obtained using a Canon EOS M50 Camera equipped with a 100 mm macro lens and a circular LED light mounted on the lens. A polarization filter mounted to eliminate reflection of light from the surface of the cervix. The camera was placed on a tripod at 30 cm distance from the surface of the cervix and was mounted on a microscope sledge to allow precise adjustment of the distance. For each patient, the microscope sledge was adjusted to ensure the orifice of the cervix was localized in the center of camera display with a free fringe around the rims of the cervix of 3–5 mm. All digital images captured during the RCT were stored in the REDCap data management system (Research Electronic Data Capture Version REDCap 9.5.6).

## Digital gridded imaging technique and cervical lesion proportion

Testing of the digital gridded imaging technique took place in the Centre for Clinical Research at North Denmark Regional Hospital after termination of the RCT. In the RCT 412 images were captured (at inclusion, at week 5, 10 and 15). Images from inclusion and week 15 were not used in the present study, since scoring of these images previously had been performed by two of the observers for another study. A portfolio of 70 digital images was established from the remaining 220 images captured at week 5 and 10. The computer software R studio (version 1.2.5033) was used to randomly select the images also allowing images from the same women (image from both visit 5 and 10) to be selected. Identifiers of the images were anonymized concerning RCT participant number and time of image capture.

The JavaScript programming software was used to develop the digital gridded imaging technique, named FGS QubiFier, for quantitative measurement of CLP based upon the identification of cervical lesions on a digital image by an observer. With the software programme, a semi-transparent circularly formed grid is superimposed as an additional layer upon the digital image of the cervix. The grid covers the whole circumference of the cervix with the orifice appearing in the center of the grid by use of a zoom technique. When the grid is correctly placed it is locked and follows the image when zoom is applied for further analysis of the image. The grid consists of 424 equally sized squares enabling the observer to mark the squares containing any of the four FGS associated lesion types manually in a structured manner, hence leaving the squares without any lesions unmarked. (Fig 1).

During the review of the 424 squares the number of squares marked by the observer is counted automatically by the FGS Qubifier software program. To obtain the final CLP measure, as a result of the review, the total number of marked squares are divided automatically by 424, equal the total number of squares in the grid, and then multiplied by a factor 100. Hence, the estimated percentage constituted the CLP in the range of zero to 100 as a quantitative measure of lesions covering the cervical portio as identified by the observer. As an example, if 25 squares of the 424 squares are marked by an observer, the CLP will then be estimated to 5.8%

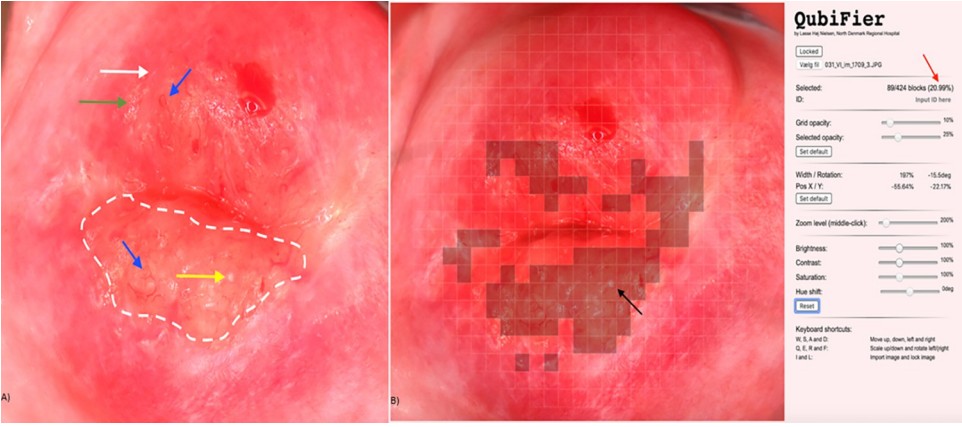

**Fig 1. Digital image showing female genital schistosomiasis associated pathognomonic lesions of the cervix (picture A) and the same lesions being digitally marked by QubiFier to determine the cervical lesion proportion (picture B).** The left picture (A) shows an area, which contains homogenous yellow sandy patches appearing as yellow colored area (indicated by white dashed line). The differently colored arrows indicate the other pathognomonic lesion types: a grainy sandy patch (white arrow), clustered grainy sandy patches or rice-grain shaped sandy patches (green arrow) and a rubbery papule colored in beige with an uneven surface (yellow arrow). Abnormal blood vessel (rounded, uneven-calibered, corkscrew or convoluted) are indicated by the blue arrow. The right picture (B) shows squares of the grid marked digitally containing any types of pathognomonic lesions. The red arrow shows the proportion of the cervix covered by any type of pathognomonic lesions.

(6.0%). Hence, measurement of CLP provides a mean for grading the cervical pathology as mild, moderate, or severe, which corresponds to intervals of low, intermediary, or high CLP measures, respectively. The number of rubbery papules are also countable by the observer and registered as a measure of rubbery papule count (RPC).

## Types of lesions associated to FGS

The following four types of lesions are occurring in FGS [14,20] and displayed in the WHO FGS pocket atlas [13].

1. *Grainy sandy patches*, defined as areas with distinct single or clusters of oblong grains (approximately $0.05 \times 0.2$ mm) in the cervicovaginal mucosa.

2. *Yellow homogenous sandy patches*, defined as homogenous foci without detectable grains at 15 times the original magnification.

3. *Rubbery papules*, previously described in the urinary bladder mucosa only, defined as spheroid, firm, beige, smooth papules (size, 0.3–1.2 mm) in the cervicovaginal mucosa.

4. *Neovascularization*, indicating pathological changes of small venules of corkscrew, unevenly calibrated and convoluted appearance [21,22]. The distorted blood vessels are the result of eggs released by a female worm which did not achieve to penetrate the wall of the blood vessel and hence become trapped in a venule, thereby blocking the natural blood flow.

## Study design

Three observers (A, B and C), all physicians trained and practicing medicine in Denmark, reviewed the 70 randomly selected images in the gridded imaging technique study (Fig 2). The observers had different levels of clinical experience regarding gynecology in general and FGS specifically: observer A (recently graduated from the medical school and just introduced to FGS at a textbook level), observer B (gynecology specialist with minor FGS field experience) and observer C (gynecology internship with major FGS field experience).

In an initial consensus rating exercise (Procedure 1), each of the three observers independently examined ten randomly selected images (images 1 to 10) and rated the cervical lesions

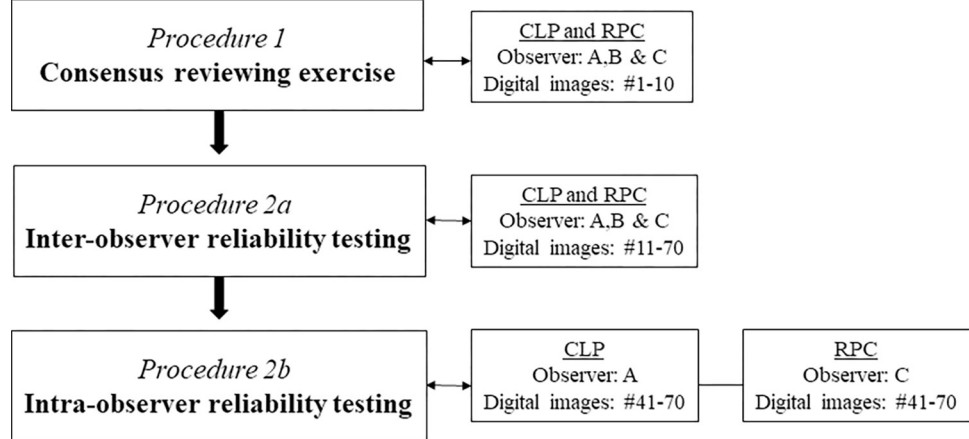

**Fig 2. Flowchart displaying the different testing procedures in the FGS digital gridded imaging technique study in which 70 images of the cervical portio were reviewed for cervical lesion proportions (CLP) and rubbery papule counts (RPC), respectively.**

(Fig 2). The three observers then shared and discussed their findings to reach consensus on the uniform rating of the images. Subsequently, the remaining 60 images (images 11 to 70) were rated independently by the three observers testing inter-observer reliability (CLP and RPC) in Procedure 2a. Two weeks later intra-rater reliability was tested for images 41 to 70 in a similarly blinded manner by observers A (CLP) and C (RPC) in Procedure 2b. To ensure sufficient scoring, the images could be reviewed at all zoom levels determined by the reviewer.

### Quality of the digital images

Unfortunately, the surface of the cervix could not always be depicted completely for anatomical reasons, e.g. when the labia majora were partially constricted or the cervix was anteverted or retroverted. The impact of the image quality on the CLP rating was therefore assessed between the three observers in separate sessions (A and B, A and C, and B and C, respectively). The quality of the digital images was stratified into whether the entire surface of the cervix was depicted optimally or partially.

### Data management and statistical analysis

The collected data were entered into REDCap data management system. Data validity was ensured by two researchers independently. Data analysis was performed by use of the RStudio programming language (version 1.2.5033).

The CLP and RP determined by the three observers was compared using the Kruskal-Wallis rank-sum test, and CLP between raters were compared using the Wilcoxon rank-sum test. To test if the gridded imaging technique were able to distinguish between clinical pathology including rubbery papules and without rubbery papules, Fleiss kappa was calculated. The value of Fleiss kappa was interpreted as follows: <0.20 (poor), 0.21 to 0.40 (fair), 0.41 to 0.60 (moderate), 0.61 to 0.80 (good) and 0.81 to 1.00 (very good) [23].

To test the degree of agreement between the three observers (inter-rater reliability) and the degree of agreement among the same observer at two timepoints (intra-rater reliability) the intraclass correlation coefficient (ICC) was calculated by a two-way mixed-effects model based on average rating values and absolute agreement. Intra-observer variation was calculated by a two-way mixed-effects model based on a single-rating and absolute agreement. The ICC values were interpreted as follows: $\leq 0.5$ (poor), 0.5 to 0.75 (moderate), 0.76 to 0.9 (good), and $> 0.90$ (excellent) [24].

## Results

### Age and parasitology

The 60 randomly selected images used for intra- and inter-observer reliability testing covered 50 different women: one image from 40 women (n = 40) obtained at either follow-up at week 5 or week 10 and two images from 10 women obtained at both follow-up visits (n = 20). Median age of the 50 women in the study was 26.5 years (interquartile range (IQR) 20.8–33.0). The median number of *S. haematobium* eggs per 50 ml of urine was 2.5 (IQR 0–62.0). In 13 women, the filtration of 50 ml urine did not reveal eggs.

### Quality of digital images

In 33 (55%) of the 60 digital images used for intra- and interobserver reliability testing, the surface of the cervix was depicted completely, whereas in 27 images (45%) up to 20% of the cervical surface was not visible. However, the observers still found the quality of images to be

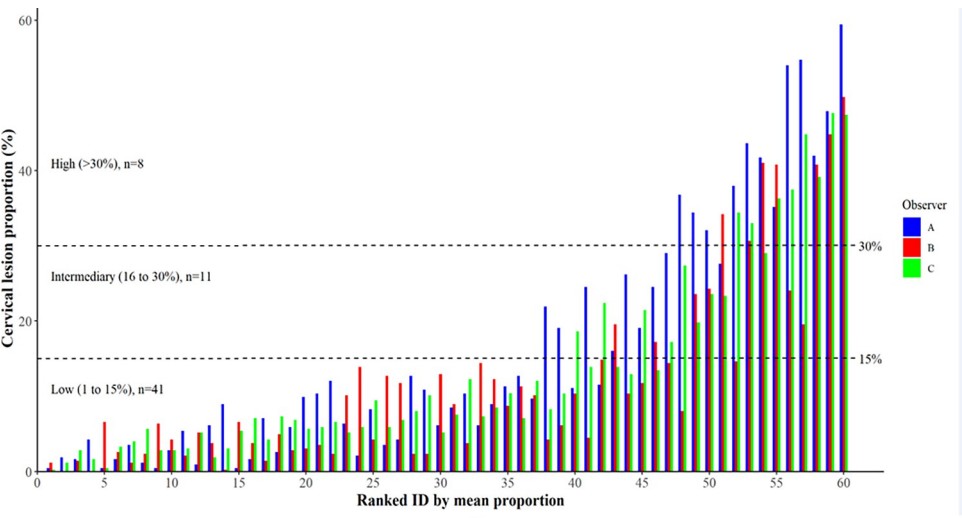

**Fig 3. Distribution of cervical lesion proportions (%) for the 60 digital images as rated by the three observers and ranked in accordance with the respective individual mean values.**

adequate for CLP and RPC determination, and all images were included in the further analysis.

## Cervical lesion proportion

In Procedure 2a (60 images were individually scored by the three reviewers), the CLP ranged from 0.5% to 59.4% in observer A, from 0% to 49.8% in observer B, and from 0.2% to 47.6% in observer C. Fig 3 shows mean CLP by the three observers for each the 60 digital images. The mean proportions were divided into three interval levels defined arbitrarily as low (1 to 15%), intermediary (16 to 30%) and high (>30%) with a distribution as follows: 68% (n = 41), 18% (n = 11) and 13% (n = 8). Three randomly selected cases representing each of the three levels are displayed in Fig 4.

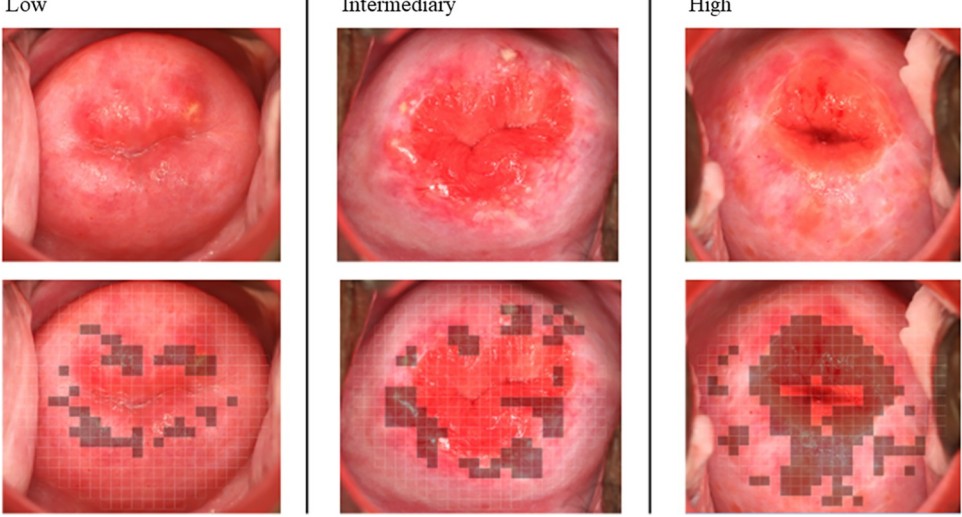

**Fig 4. Three cases representing different levels of female genital schistosomiasis (FGS) associated pathology in accordance with the cervical lesion proportion (CLP) categories: low (1 to 15%), intermediary (16 to 30%), and high (>30%).**

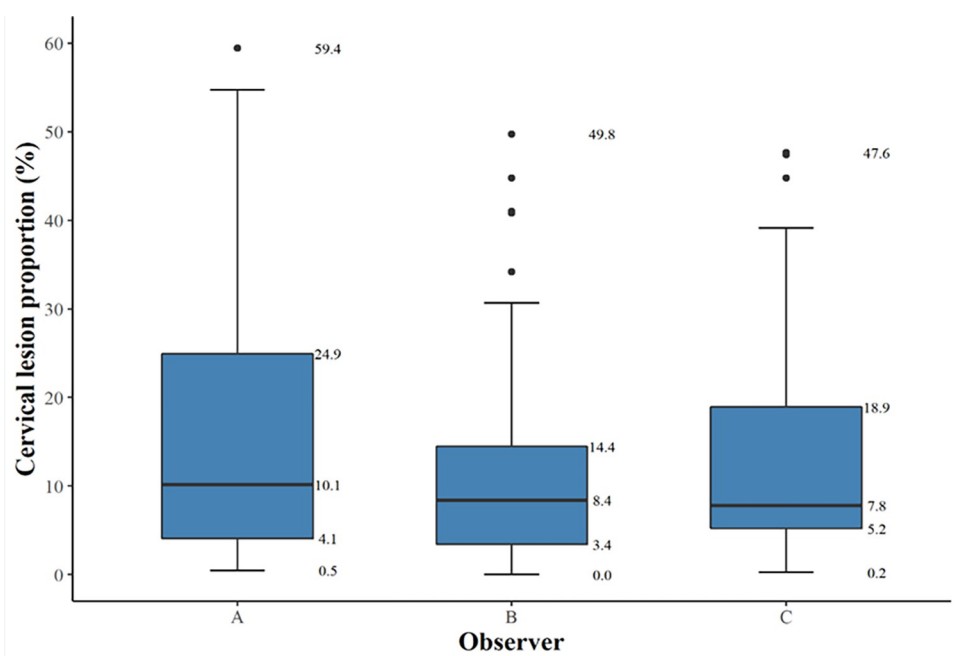

**Fig 5.** A box-and-whisker-plot of cervical lesion proportions (%) for each observer (A, B, and C) rating the 60 digital images. The bold horizontal line indicates the median, the upper and the lower line of the box the interquartile range.

Median CLP (interquartile range, IQR) for observers A, B and C was 10.1% (4.1–24.9), 8.4% (3.4–14.5) and 7.8% (5.2–18.9), respectively (Fig 5). The difference in scores between all raters was not statistically significant (p = 0.094). Hence, inter-rater differences were not significant between the raters; A vs. B p = 0.53, A vs. C p = 0.53 and B vs. C p = 0.65. The scoring over time between the same reviewer (Procedure 2a versus Procedure 2b) the median CLP (IQR) was 6.3% (1.7–11.9) in Procedure 2a and 8.1% (3.4–14.0) in Procedure 2b. The difference was not statistically significant (p = 0.294).

## Rubbery papule count

In Procedure 2a, the RPC ranged from 0 to 18 in observer A, from 0 to 19 in observer B, and from 0 to 28 in observer C. Fig 6 shows the distribution of RPC for each of the 60 digital images. Median RPC (IQR) for observers A, B and C were 1 (0–3), 2 (0–4) and 2 (0–4), respectively (Fig 7). Difference in scores between all raters was not statistically significant (p = 0.341). Inter-rater differences were not significant; A vs. B p = 0.41, A vs. C p = 0.41 and B vs. C p = 0.97. Median RPC (IQR) was 1 (0–3) in Procedure 2a and 1 (0–2) in Procedure 2b. The difference was not significant (p = 0.055). The Fleiss kappa value for the three observers in distinguishing rubbery papules was 0.55 (IC 0.54–0.55; p<0001), demonstrating a moderate agreement.

## Inter- and intra-observer reliability

The inter-observer reliability (A, B and C) for CLP measured by ICC was 0.93 (95% CI 0.90–0.96) indicating an excellent performance. The inter-observer reliability (A, B and C) for RPC measured by ICC was 0.88 (95% CI 0.82–0.92), indicating a good performance. The intra-observer reliability (A) for CLP measured by the intra-class correlation coefficient ICC was 0.90 (95% CI 0.79–0.95), indicating an excellent performance. The intra-observer reliability (C) for RPC measured by ICC was 0.80 (95% CI 0.59–0.90), indicating a good performance.

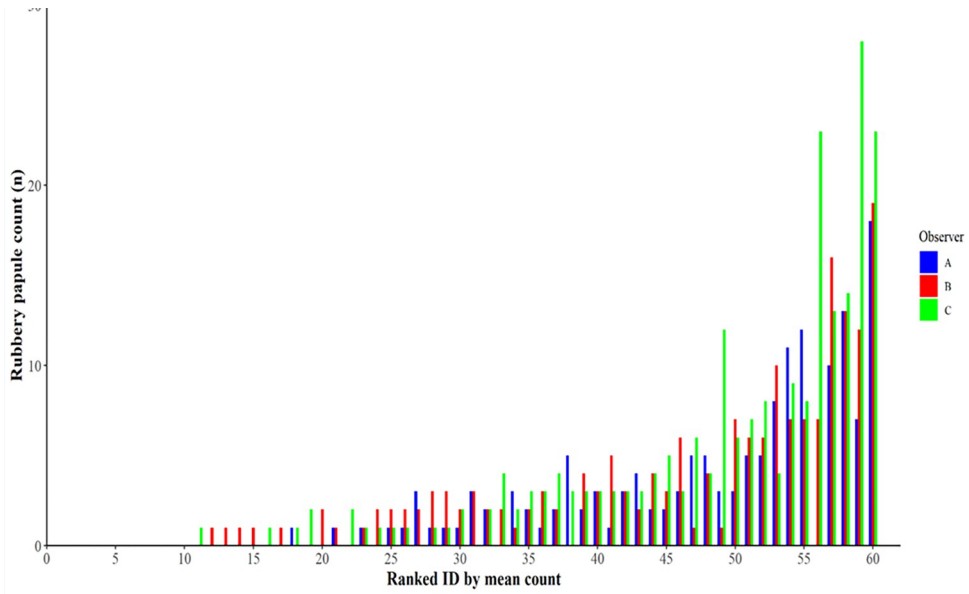

**Fig 6.** A box-and-whisker plot of median rubbery papule counts (n) and interquartile range for each observer (A, B, and C) rating the 60 digital images.

### Factors influencing discrepancies

Data were further analyzed to identify whether certain characteristics of the surface of the cervix contributed to significant discrepancies in CLP between observers. In ten images with the most pronounced discrepancy in CLP between observers A and B, the differences ranged from 13% to 35%. When comparing observer A with observer C, the differences ranged from 11% to 17% for the ten images representing the most pronounced discrepancy between the two,

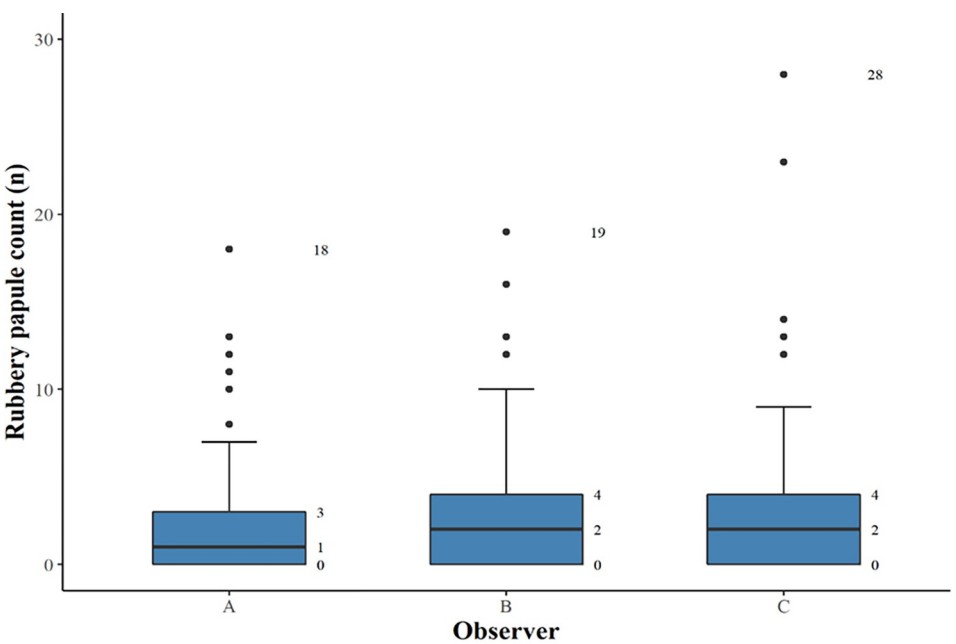

**Fig 7. Distribution of rubbery papule count (n) for the 60 digital images as rated by the three observers and ranked in accordance with the respective individual mean values.**

and for observers B and C, differences ranged between 8% to 25%. Two major domains contributed to discrepancies between observers: 1. difference in zoom-level of the digital images resulting in a different coverage of the surface of the cervix by the grid, 2. disagreement when different cervical lesions co-existed and overlapped in the same square particularly in case of neovascularization. In addition, disagreement occurred also in a few cases due to reduced image quality and/or because the glandular lining of the endocervix was intense and categorized as yellow homogenous sandy patches.

## Discussion

In this study, we have developed and tested a digital gridded imaging technique for assessment of cervical lesions occurring in FGS. Using this technique, we demonstrated that CLP and RPC are useful measures enabling quantification of cervical pathology. The four different lesion types (grainy sandy patches, yellow homogeneous sandy patches, rubbery papules, and neovascularization) searched for by the observers in the review of the digital images may not all be associated solely to FGS. Sandy patches and neovascularization are also associated to other conditions commonly affecting women living in *Schistosoma* endemic areas, including sexually transmitted infections (e.g. human papilloma virus and herpes simplex virus), likewise rubbery papules may resemble nabothian cysts [15]. In any case, the technique provides an opportunity to explore further into the natural history of FGS from a clinical as well as an epidemiological perspective. Moreover, the CLP and RPC measures offer an opportunity to assess pathological changes following treatment of FGS, in addition to measurement of subsequent re-occurrence of pathology after repeated exposure to *Schistosoma* infested water.

The proposed arbitrary grading of cervical pathology as low (1 to 15%), intermediary (16% to 30%) and high (>30%) according to the CLP, requires further evaluation in other groups of women with FGS. The CLP cut-off points were arbitrarily selected for purpose of the study, and these should be tested clinically to investigate if different levels of cervical pathology correlate with genital morbidity, including gynecological complaints.

The ICC values for the CLP and RPC indicated good to excellent inter- and intra-observer reliability. The difference in inter-observer reliability concerning CLP (excellent) and RPC (good) can be explained by the difficulty in evaluating a pronounced convex structure such as a rubbery papule on a 2D surface [15]. A moderate agreement between the three observers was found regarding identification of rubbery papule lesions, emphasizing another important aspect in the assessment of the gridded imaging technique. The kappa value demonstrated that the observers were capable of distinguishing between the presence versus absence of this FGS associated lesion. We believe that rubbery papule lesions represent the sign of recent eosinophil inflammatory response compared to the sandy patch lesions representing a late stage in the natural history of FGS. Thus, a validated RPC could be a good proxy for recent deposition of eggs in the epithelium.

The comparison of CLP and RPC median scores between observers showed no significant difference between any of the observers but for a few images a discrepancy up to 35% was noted. Two major domains were identified when evaluating the cause of discrepancy between observers. Firstly, different zoom-levels between the observers, and secondly, co-existence of lesions in same grid square. Reduced image quality also played a role in the discrepancy between observers. The possible causes of discrepancies between observers should be addressed in future studies to further optimize the gridded imaging technique protocol.

Use of digital images for the diagnosis and management of FGS provides different advantages in comparison to the colposcopy. FGS associated lesions are more easily identified on a computer screen as it is possible to enlarge portions of the cervix by increasing the zoom level,

thus providing a more detailed visualization of the lesions. Digital images can be revaluated any time and shared with other observers or health care professionals for a second clinical opinion. The digital images of the cervix can be captured by use of the camera alone or by using a camera in conjunction with the colposcopy. A camera with proper technical features may potentially replace the colposcope. Introduction and use of the colposcope in FGS control programs is challenged by lack of supporting health care and staff training capacity as well as logistic and financial constraints. Previous studies by Holmen and colleagues have suggested use of colorimetric image analysis as a diagnostic tool in FGS [17,25]. However, the computer analysis is performed on colposcopy images, which are typically not available in district hospitals in sub-Saharan Africa. The digital camera image documentation approach overcomes the shortage of colposcopy equipment. However, access to cameras may also be limited in rural high endemic areas. Previous studies on visualization of cervical pre-cancer pathology have shown that image documentation could be obtained by mobile phone cameras [26,27], which seems also to have a major potential in a large-scale FGS diagnostic context.

## Limitations

This study has some limitations. Firstly, underestimation of the CLS and RPC may have occurred because up to 20% of the cervical surface was not visible in half of the images selected for this study. Secondly, colposcopy is currently considered the standard procedure for identification of FGS associated lesions. In this study, the digital images used for the development and validation of the gridded imaging technique were obtained in women living in an *S. haematobium* endemic area enrolled in a therapeutic trial prior to this study. Main inclusion criteria for participation in this trial were presence of FGS associated lesions. However, the lesions were identified by digital camera images and not by colposcopy. A combined use of a colposcope and digital camera could have made it possible to study important comparative findings between the two diagnostic methods. As another limitation in this study, digital gridded imaging technique was not compared with real-time *PCR* for the detection of *Schistosoma* DNA in genital specimens as an accepted FGS diagnostic method. Moreover, other potential causes of homogeneous yellow sandy patches and neovascularization, such sexually transmitted infections (e.g. *Chlamydia trachomatis*, herpes simplex virus and human papilloma virus) and malignancy, respectively, were not included in a comparative analysis.

## Conclusion

The digital gridded imaging technique provides an opportunity to quantify FGS associated cervical lesions. Furthermore, we found an excellent reliability for the CLP and a good reliability for RPC. Future studies need to explore to what extent *Schistosoma* DNA PCR as an FGS diagnostic biomarker and severity of disease as perceived by the patient correlate with CLP and RPC. If so, the digital gridded imaging technique provides new opportunities in future FGS research and control programs. The study has shown that the WHO FGS pocket atlas constitutes a very useful reference guide for physicians to identify FGS associated cervical lesions regardless of clinical experience level. Use of WHO FGS atlas together with the digital gridded imaging technique, allows for potential deployment in *S. haematobium* endemic countries in conjunction with future epidemiological and therapeutic studies aiming at improved FGS control in Africa.

## Acknowledgments

We like to thank the field team for their dedicated work and the women in the district of Ambanja in the Northwest region of Madagascar for willing to participate in the study.

Photographer Lasse Høj Nielsen contributed with expertise in development of the gridded clinical image technique,

## Author Contributions

**Conceptualization:** Louise Thomsen Schmidt Arenholt, Peter Derek Christian Leutscher.

**Data curation:** Louise Thomsen Schmidt Arenholt, Katrina Kaestel Aaroe, Kanutte Norderud, Bodo Sahondra Randrianasolo, Charles Emile Ramarokoto, Oliva Rabozakandraina, Dorthe Broennum, Peter Derek Christian Leutscher.

**Formal analysis:** Louise Thomsen Schmidt Arenholt, Mads Lumholdt, Dorthe Broennum, Peter Derek Christian Leutscher.

**Funding acquisition:** Hermann Feldmeier, Peter Derek Christian Leutscher.

**Investigation:** Peter Derek Christian Leutscher.

**Methodology:** Louise Thomsen Schmidt Arenholt, Mads Lumholdt, Peter Derek Christian Leutscher.

**Project administration:** Bodo Sahondra Randrianasolo, Charles Emile Ramarokoto, Oliva Rabozakandraina, Dorthe Broennum, Peter Derek Christian Leutscher.

**Resources:** Bodo Sahondra Randrianasolo, Charles Emile Ramarokoto, Oliva Rabozakandraina, Peter Derek Christian Leutscher.

**Software:** Louise Thomsen Schmidt Arenholt, Mads Lumholdt, Dorthe Broennum, Hermann Feldmeier, Peter Derek Christian Leutscher.

**Supervision:** Bodo Sahondra Randrianasolo, Dorthe Broennum, Peter Derek Christian Leutscher.

**Validation:** Bodo Sahondra Randrianasolo, Charles Emile Ramarokoto, Oliva Rabozakandraina, Dorthe Broennum.

**Writing – original draft:** Louise Thomsen Schmidt Arenholt, Katrina Kaestel Aaroe, Kanutte Norderud, Peter Derek Christian Leutscher.

**Writing – review & editing:** Louise Thomsen Schmidt Arenholt, Mads Lumholdt, Bodo Sahondra Randrianasolo, Dorthe Broennum, Hermann Feldmeier, Peter Derek Christian Leutscher.

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
