## [Decision Letter · Decision Letter 0]

22 Jan 2022

Dear Prof Leutscher,

Thank you very much for submitting your manuscript "Validation of cervical lesion proportion measure using a gridded imaging techique to assess cervical pathology in women with genital schistosomiasis" for consideration at PLOS Neglected Tropical Diseases. As with all papers reviewed by the journal, your manuscript was reviewed by members of the editorial board and by several independent reviewers. In light of the reviews (below this email), we would like to invite the resubmission of a significantly-revised version that takes into account the reviewers' comments. 

We thank the authors for this important work, and agree that there is an urgent need to quantify FGS lesions. Please address reviewers' comments in order to strengthen this manuscript, which has some significant weaknesses at present. The manuscript could be made clearer particularly regarding whether the computer or the observers performed the actual counting, and precision of language throughout, as noted by Reviewer 1, is needed.

It will also be important to describe (1) rates and types of STIs that could confound this analysis, (2) the non-specificity of lesions (particularly neovascularization and homogeneous sandy patches), and to report the frequencies of the four different WHO atlas findings.

Finally, please double-check the whole text for typos and editing mistakes.

We cannot make any decision about publication until we have seen the revised manuscript and your response to the reviewers' comments. Your revised manuscript is also likely to be sent to reviewers for further evaluation.

Sincerely,

Jennifer A. Downs, M.D., Ph.D.

Associate Editor

Francesca Tamarozzi

Deputy Editor

We thank the authors for this important work, and agree that there is an urgent need to quantify FGS lesions. Please address reviewers' comments in order to strengthen this manuscript, which has some significant weaknesses at present. The manuscript could be made clearer particularly regarding whether the computer or the observers performed the actual counting, and precision of language throughout, as noted by Reviewer 1, is needed.

It will also be important to describe (1) rates and types of STIs that could confound this analysis, (2) the non-specificity of lesions (particularly neovascularization and homogeneous sandy patches), and to report the frequencies of the four different WHO atlas findings.

Reviewer's Responses to Questions

**Key Review Criteria Required for Acceptance?**

**Methods**

-Are the objectives of the study clearly articulated with a clear testable hypothesis stated?

-Is the study design appropriate to address the stated objectives?

-Is the population clearly described and appropriate for the hypothesis being tested?

-Is the sample size sufficient to ensure adequate power to address the hypothesis being tested?

-Were correct statistical analysis used to support conclusions?

-Are there concerns about ethical or regulatory requirements being met?

Reviewer #1: Major Comments:

1) Validation: I believe a validation study would compare a new technology against a “gold standard” to confirm that the new technology accurately measures what it claims to measure. Admittedly, there is not a consensus reference standard for FGS. However, was there a comparison made with another accepted diagnostic method for FGS (biopsy, PCR, etc)? If there was no comparison with PCR or biopsy (or Holmen’s colorimetric image analysis), I’m not sure it is correct to call this a validation study. Please rephrase the title and line 301. Please also acknowledge this in the limitations. 

2) What proportion of women in the study had STI? Please see PMID: 15772328. In Kjetland 2005, FGS-associated lesions (the homogeneous yellow sandy patch) were associated with STI (please see Table 4 in Kjetland 2005). There was strong evidence that chlamydia, HSV, and high risk HPV were associated with the homogeneous yellow sandy patches. Thus, to ensure that the lesions described were not associated with STI, please describe what STI testing was performed and what proportion of the women with sandy patches also had STI.

3) Please could the authors describe any assessment for dysplasia or malignancy? Neovascularization (see Kjetland 2005 Table 5) has been strongly associated with CIN I-III). Thus, in lines 114 & 116 these sentences could say “in women with FGS-associated lesions” or some other language to inform the reader that we do not know with certainty that these lesions are due to FGS, rather only that they look like lesions that are associated with FGS. 

4) Given that FGS lesions lack specificity, I think it is challenging to call these “pathognomonic” lesions. Please rephrase throughout. Additionally, without an evaluation for STI + malignancy + other diagnostic methods for FGS, it seems appropriate to use less definitive language around causality of these lesions by FGS (see lines 36 (fgs-evoked), line 42-43 “women with FGS”, line 116 “women with FGS”, line 130). However, in some locations the authors are examples of more measured language, suggesting association rather than causality. For example, at line 35 the authors say that “Female genital schistosomiasis (FGS) is characterized by a pattern of lesions”. I think this type of language throughout would be more appropriate. 

5) Abstract: (Line 38) You say that “no tools are able to quantify clinical pathology at the cervix in a standardized and reproducible manner”. However, in the discussion you reference Holmen’s colorimetric image analysis. Is that tool not standardized and reproducible? Also Holmen’s method can also be applied to digital images.

6) Methods:

a. Line 159 – 164. It seems to me from these lines that the computer algorithm automatically counted and marked the FGS associated lesions (so the computer algorithm calculates the CLP)? If this is true, what was the role of the human reviewers? In line 214 it appears the reviewers also calculate a CLP – how is this done? Without a clear understanding of the human component, it is challenging to understand how meaningful the inter-rater reliability is.

b. Lines 165-167 – to clarify, if there were 25 lesions/squares, this number 25/424 = 6% coverage of the cervical surface? These sentences might need to be rephrased slightly - this information was challenging to access.

c. Line 188 - What are the two components of the experience profile (1+1) – is one number for gynecology and one for FGS? This should be clarified, and how does one ‘achieve’ minor/moderate/major experience? This seems slightly arbitrary and should be further explained and disclosed as such (as you did very nicely with the grading of lesions into mild/mod/severe).

Minor Comments:

1) Lines 178 - Types of lesions. I think there is some debate regarding whether neovascularization should be included as an FGS associated lesion (in PMID: 25412334 the authors note that only homogeneous and grainy sandy patches, along with rubbery papules meet the criteria for FGS based the result of an expert meeting in 2010). I acknowledge that all 4 FGS-associated lesion the authors describe are displayed in the pocket atlas, thus making all 4 reasonable to include. However, it might be nice to have a sentence in the discussion outlining the controversy. 

2) Referencing

a. Line 82 – epidemiology (Africa and the middle east) – please reference

b. Line 83 – how does Shaematobium affect the genital organs? Please expand and reference.

c. Line 64 – please reference these complaints. There is only one study I am aware of in which dyspareunia is associated with FGS (PMID: 21572820)

d. Line 66 (Author Summary). I think it is an overstatement to say that “FGS is considered as a major risk factor for transmission of HIV” – please rephrase 

e. Line 89. I think it is an overstatement to say “FGS associated with increased risk of HIV acquisition”. There has only been one study suggesting an association with HIV acquisition and this was a retrospective study in women with Shaematobium antibodies (they did not have FGS evaluations) (PMID 30543654). Please rephrase.

f. Line 90 – While it is a compelling hypothesis, I am unsure that we have evidence to claim that FGS increases the risk of HPV infection. The references you have cited do not support this statement (#9-11). Please amend this statement or provide revised references.

g. Line 87-88. Please elaborate on which concrete symptoms or features of FGS (?dyspareunia, discharge, genital itching) might lead to a “poor quality of sexual life”.

3) Other minor comments:

Line 38 (abstract): “A tool for quantification of the lesions is needed to improve FGS research and control programs” – would the tool also be useful for evaluating treatment?

Line 70-72 – How does your tool allow for treatment at the community level? Would your tool also allow for monitoring of lesion resolution (as well as lesion development)?

Line 122 – should RCT stand for “randomized controlled trial”?

Line 127 – do you mean that the area highly endemic for S. haematobium (with a high prevalence of FGS?)?

Line 142 – what is the digital part of the study – do you mean the image review?

Line 346 – would there be a role for a mobile phone colposcope here as well?

Reviewer #2: The objectives are clear: This study aimed to develop and validate a cervical lesion proportion (CLP) measure for quantification of cervical pathology in FGS. 

The RCT study design has been clearly explained.

The sample selected from the population needs further explanation. There were 116 women enrolled in the RCT between 15-35y with FGS pathognomonic lesions. How were 70 of the 412 images selected? For the comparison between the observers, why were 40 images selected from weeks 5 or 10 and two images from each of 10 women at both visits? Which women were selected, and why?

**Results**

-Does the analysis presented match the analysis plan?

-Are the results clearly and completely presented?

-Are the figures (Tables, Images) of sufficient quality for clarity?

Reviewer #1: 1) Results:

a. What proportion of each of the FGS-associated lesions did the authors find? This is important given that neovascularization is not uniformly considered to be FGS-associated.

b. Please review for the reader again here which steps were 2a and 2b.

c. It might be helpful to define inter-observer and intra-observer reliability for those readers who do not have a background with these methods

Reviewer #2: The results are accompanied by figures showing the comparisons of the findings by the three different observers.

Please can the authors further elucidate the proportion of homogeneous sandy patches and grainy sandy patches found.

Since getting photographs of the cervix may be difficult it would be on interest to know if when only 20% pf the cervical surface was only partially depicted, when the similar results were found between the more and less experienced observers. Can the authors please comment?

**Conclusions**

-Are the conclusions supported by the data presented?

-Are the limitations of analysis clearly described?

-Do the authors discuss how these data can be helpful to advance our understanding of the topic under study?

-Is public health relevance addressed?

Reviewer #1: -I'm afraid from the way the methods are written that I don't fully understand the role of the human reviewers if the computer algorithm is calculating the CLP (please see above).

-Please see above for additional items for the limitations

Reviewer #2: The conclusions are supported by the data and they describe the limitations, but since a colposcope is the recommended tool, the authors need to consider a way forward. 

The authors do not suggest a comparison study between use of the the colposcope (a heavy and expensive piece of equipment) and the digital image documentation approach which overcomes the shortage of colposcopic equipment. We would need information comparing the accuracy of the colposcope and the camera, to determine the accuracy/effectiveness of this new approach.

Finding an effective solution to the diagnosis of FGS urgently needs such a study .

**Editorial and Data Presentation Modifications?**

Reviewer #1: 1) Line 190-199 – The authors have explained that the 70 images were chosen randomly – was this from a total of 412 total images from the study? Overall, I find it the numbers (70 from 212 or 412) and timepoints (the majority from week 5 but some from week 10 but from 50 women) hard to follow. Taking one randomly selected image per woman makes sense (line 198) but why then 2 from 10 women? This needs clearer explanation and perhaps Figure 2 needs a flow diagram to show the total number of participants, the time points, and images. This would allow further understanding of the risk of bias. The explanation given from lines 237-239 is easier to follow than the explanation used in the methods.

2) Reducing stigma – I think we as scientists can do our part to reduce stigma through the language we choose in our manuscripts. Please consider rephrasing:

a. Line 93 “resource poor”

b. Line 102 “in rural Africa”

c. Line 308 “contaminated” water 

d. Line 346 “in an African context”

Reviewer #2: (No Response)

**Summary and General Comments**

Reviewer #1: This important study evaluates the use of a clinical lesion proportion measure for quantification of cervical pathology in Madagascan women with FGS. The clinical lesion proportion allowed a reproducible quantification of the surface of the cervix affected by FGS lesions. I agree with the authors that a quantitative and reproducible means to evaluate FGS-associated cervical lesions would be beneficial to the field. I also agree with the authors that a quantifiable tool would improve research, control and also treatment. I also agree that having a quantifiable tool will be useful in investigating whether certain levels of genital pathology correlate with symptoms.

Reviewer #2: (No Response)

PLOS authors have the option to publish the peer review history of their article (what does this mean?). If published, this will include your full peer review and any attached files.

Reviewer #1: No

Reviewer #2: No
---

## [Decision Letter · Decision Letter 1]

2 Apr 2022

Dear Prof Leutscher,

Thank you very much for submitting your manuscript "Cervical lesion proportion measure using a digital gridded imaging techique to assess cervical pathology in women with genital schistosomiasis" for consideration at PLOS Neglected Tropical Diseases. As with all papers reviewed by the journal, your manuscript was reviewed by members of the editorial board and by several independent reviewers. The reviewers appreciated the attention to an important topic. Based on the reviews, we are likely to accept this manuscript for publication, providing that you modify the manuscript according to the review recommendations. 

Thank you for your responsiveness to the prior reviews. The manuscript is substantially improved. However, there are still some items that need to be revised before this manuscript can be accepted. These are outlined by the reviewer. Those most critical for revision include:

1. Please improve image resolution for the figures, which are currently illegible.

2. Please be more precise in the use of the word "pathognomonic," which still remains in places in the text. This should either be removed, or should be better justified as pathognomonic based on other studies.

3. Please revise statements about FGS being a possible risk factor for HIV transmission so that they are more circumspect, as there are not sufficient data to conclude this strongly.

I believe that the use of the Wilcoxon rank-sum test to compare median CLP and median numbers of rubbery papules in the Methods section is acceptable and does not need further revision.

Sincerely,

Jennifer A. Downs, M.D., Ph.D.

Associate Editor

Francesca Tamarozzi

Deputy Editor

Thank you for your responsiveness to the prior reviews. The manuscript is substantially improved. However, there are still some items that need to be revised before this manuscript can be accepted. These are outlined by the reviewer. Those most critical for revision include:

1. Please improve image resolution for the figures, which are currently illegible.

2. Please be more precise in the use of the word "pathognomonic," which still remains in places in the text. This should either be removed, or should be better justified as pathognomonic based on other studies.

3. Please revise statements about FGS being a possible risk factor for HIV transmission so that they are more circumspect, as there are not sufficient data to conclude this strongly.

I believe that the use of the Wilcoxon rank-sum test to compare median CLP and median numbers of rubbery papules in the Methods section is acceptable and does not need further revision.

Reviewer's Responses to Questions

**Key Review Criteria Required for Acceptance?**

**Methods**

-Are the objectives of the study clearly articulated with a clear testable hypothesis stated?

-Is the study design appropriate to address the stated objectives?

-Is the population clearly described and appropriate for the hypothesis being tested?

-Is the sample size sufficient to ensure adequate power to address the hypothesis being tested?

-Were correct statistical analysis used to support conclusions?

-Are there concerns about ethical or regulatory requirements being met?

Reviewer #1: This important study has been very nicely revised. Technology that allows the visual findings associated with FGS to be quantified and reproduced is an important contribution to the field. This work evaluates the use of a clinical lesion proportion measure for quantification of cervical pathology in Madagascan women with FGS. The clinical lesion proportion allowed a reproducible quantification of the surface of the cervix affected by FGS lesions. I agree with the authors that a quantifiable tool would improve research, control and also treatment. I also agree that having a quantifiable tool will be useful in investigating whether certain levels of genital pathology correlate with symptoms and other diagnostics. This work is needed will have an impact on the field.

I do have a few questions about the methods and statistics:

1) Line 288 – 290. It seems interesting that the mean RPC count was statistically different between reviewers A and B and also A and C for the RPC count when the numbers were so small (0-3, 0-3, and 0-4). Since these numbers are so low, is there another way to compare them that would be more meaningful?

2) Line 291-292. I am not sure how the median RPC could be significantly different in step 2a and 2b can be different when they are both 1? “Median RPC (IQR) was 1 (0-3) in Step 2a and 1 (0-2) in Step 2b, thus statistically significant (p=0.002)”

3) Lines 218-223: When the impact of the image quality on the CLP rating was assessed between the reviewers in separate sessions , the authors say that the images were stratified to whether they were depicted optimally or suboptimally? I see later in line 264 – 265 that the reviewers felt that the reviewers felt that the quality of all the images was adequate for CLP and RPC determination? So could I clarify that ultimately no images were stratified to the group that was suboptimally depicted?

4) Could I clarify that while all observers had access to the same images, each observer could choose to rate/view the images at different zoom settings?

**Results**

-Does the analysis presented match the analysis plan?

-Are the results clearly and completely presented?

-Are the figures (Tables, Images) of sufficient quality for clarity?

Reviewer #1: The Figures are a nice addition, but Figure 2 and 3 were challenging to access due to the text size

1) Figure 1 – This is a nice figure. Unfortunately the text under the “qubifier” heading is too small to read

2) Figure 3 – Unfortunately the labeling of the axes is too small and blurry to read

**Conclusions**

-Are the conclusions supported by the data presented?

-Are the limitations of analysis clearly described?

-Do the authors discuss how these data can be helpful to advance our understanding of the topic under study?

-Is public health relevance addressed?

Reviewer #1: Yes, overall the conclusions are supported by the data presented. The limitations of the analysis are clearly described. The authors discuss how the data can be helpful to advance the field of FGS.

I do still have concerns about how two of the FGS lesions are still presented in the conclusion as being pathognomonic, without adequate data to support this conclusion. Given that FGS lesions lack specificity, I think it is challenging to call any of the four lesions “pathognomonic” (even rubbery papules or grainy sandy patches). 

For example, in the discussion line – 326 – 327 the authors say “it seems reasonable to categorize grainy sandy patches and rubbery papules as FGS pathognomonic lesions.” Where in the manuscript have the authors provided evidence to support this statement? Since rubbery papules can look like Nabothian cysts, thus I think this position will be challenging to support (I do acknowledge that in Randrianosolo 2015 that 100% of the rubbery papules biopsied contained eggs [Table 2], but this is a small number). Additionally, in Kjetland’s 2005 “Simple clinical manifestations” paper, if I have read the tables correctly, didn’t only 30-60% of women with grainy sandy patches have eggs in any genital specimen vs biopsy (Table 3)? I am not sure this evidence is compelling enough to consider the grainy sandy patches pathognomonic either. Please consider either removing this statement, or providing additional evidence. If there is compelling evidence to cite, please could the authors consider adding this at line 185 where the four lesions are described.

I have still found a few other instances where FGS lesions are referred to as “pathognomonic”. 

b. Discussion line 345 – 346 In this sentence the authors describe the rubbery papule as a “fgs pathognomonic lesion”. Please rephrase.

c. Line 133 – In the inclusion criteria – there is a reference to pathognomonic FGS lesions, please rephrase.

d. Line 185 – "types of pathognomonic lesions" please rename this section to be congruent with the changes made in the last revisions (Types of FGS associated cervical changes/lesions etc)

**Editorial and Data Presentation Modifications?**

Reviewer #1: 1) Line 46 - one type of FGS associated lesions (Please check plurals, should this be lesion?)

2) Line 51-53 - I think the last few sentences of the abstract might need to be clarified. Currently, the way the abstract is worded (in regards to the natural history of egg-induced pathology and assessing praziquantel treatment efficacy) it almost seems like these measures will be evaluated in this manuscript. Perhaps adjusting the wording to clarify this will be a future endeavor would be helpful here, such as: “In the future, grading of cervical pathology by CLP may provide insight into the natural course of schistosome egg-induced pathology of the cervix and may have a role in assessing praziquantel treatment efficacy against FGS.”

3) Lines 62- 74 - Author Summary: This segment gives a nice summary about FGS and the future uses of this technology but I am not sure that this segment tells the reader what the manuscript is about. I think the main aim was “The aim of the study was to test inter- and intra-rater reliability using a cervical lesion proportion (CLP) measure in a series of digital images from women living in a Schistosoma haematobium hyperendemic area in Madagascar”. Perhaps this could be brought into the author summary? 

4) Lines 85 – 89. The content of these sentences is challenging to access because the sentence is so long. Could it potentially be broken up with a period (potentially between the “anastomoses in the pelvic region” and “eggs”?

5) Please review lines 88 and 92 for sentence fragments.

6) Line 92-93. Could the sentence containing the “poor quality of sexual life” potentially be rephrased to express a more objective concept? I think the authors are saying that FGS may have a negative impact on the woman’s sexual experience, but the current phrasing feels somewhat subjective. The authors could also consider removing this phrase.

7) Schistosoma should be italicized throughout – please review lines 98, 324, 331, 393, 402

8) Line 113 – “imperatively needed” feels redundant to me – could one word or the other suffice?

9) Line 167: superimposed?

10) Line 344: validation (Since this is not a validation study, do the authors mean assessment?)

11) Lines 323-326 – please provide a reference.

12) Lines 378-379: I’m not sure what you mean by “as mobile phones with camera features have pioneered the African continent” and I’m wondering if this particular portion of the sentence is necessary? And if necessary, should this be referenced?

**Summary and General Comments**

Reviewer #1: The revised manuscript is much clearer and is easier to follow. Thank you for the helpful revisions. 

I do think that the wording around the association between FGS and HIV could be clarified by the authors. I believe there has only been one study suggesting an association with HIV transmission and schistosomiasis in women and this was a retrospective study in participants with Schistosoma antibodies (the participants in this study did not have FGS evaluations) (PMID 30543654). Thus, I think it is not accurate to state that FGS is considered a risk factor for HIV transmission.

I think it is also important to be clear regarding the difference between the association between schistosomiasis/FGS and HIV. At line 94, the authors reference a meta-analysis by Patel et al. The abstract for the Patel article clearly says “A significant association of schistosomiasis with HIV was found, however, a specific summary estimate for FGS could not be generated.” If an association with HIV is discussed, based on the Patel reference (12), I am not sure that it is correct to make the leap to FGS and then again to HIV transmission. Reference 11 is a hypothesis piece and can't stand alone to support the association between FGS and HIV transmission. 

a. Line 66 (Author Summary). I think it is still an overstatement to say that “FGS is considered as a risk factor for transmission of HIV” – please rephrase. 

b. Line 94. I still think it is an overstatement to say “FGS is suspected to be contributing risk factor for transmission of human immunodeficiency virus (HIV) in Schistosoma endemic areas of Sub-Saharan Africa”. Please rephrase.

PLOS authors have the option to publish the peer review history of their article (what does this mean?). If published, this will include your full peer review and any attached files.

Reviewer #1: No

Figure Files:

Data Requirements:

Reproducibility:

References

---

## [Editor Report · Decision Letter 2]

19 May 2022

Dear Prof Leutscher,

We are pleased to inform you that your manuscript 'Cervical lesion proportion measure using a digital gridded imaging techique to assess cervical pathology in women with genital schistosomiasis' has been provisionally accepted for publication in PLOS Neglected Tropical Diseases.

Best regards,

Jennifer A. Downs, M.D., Ph.D.

Associate Editor

Francesca Tamarozzi

Deputy Editor

Please note that the figures are still blurry. Please be sure to improve resolution for publication.

---

## [Editor Report · Acceptance letter]

20 Jun 2022

Dear Prof Leutscher,

We are delighted to inform you that your manuscript, "Cervical lesion proportion measure using a digital gridded imaging techique to assess cervical pathology in women with genital schistosomiasis," has been formally accepted for publication in PLOS Neglected Tropical Diseases.

Best regards,

Shaden Kamhawi

co-Editor-in-Chief

Paul Brindley

co-Editor-in-Chief
